# Low-Complexity Nonparametric Bayesian Online Prediction with Universal Guarantees

**Alix Lhéritier**
Amadeus SAS
F-06902 Sophia-Antipolis, France
`alix.lheritier@amadeus.com`

**Frédéric Cazals**
Université Côte d'Azur
Inria
F-06902 Sophia-Antipolis, France
`frederic.cazals@inria.fr`

## Abstract

We propose a novel nonparametric online predictor for discrete labels conditioned on multivariate continuous features. The predictor is based on a feature space discretization induced by a full-fledged k-d tree with randomly picked directions and a recursive Bayesian distribution, which allows to automatically learn the most relevant feature scales characterizing the conditional distribution. We prove its pointwise universality, i.e., it achieves a normalized log loss performance asymptotically as good as the true conditional entropy of the labels given the features. The time complexity to process the $n$-th sample point is $O(\log n)$ in probability with respect to the distribution generating the data points, whereas other exact nonparametric methods require to process all past observations. Experiments on challenging datasets show the computational and statistical efficiency of our algorithm in comparison to standard and state-of-the-art methods.

## 1 Introduction

**Universal online predictors.** An *online (or sequential) probability predictor* processes sequentially input symbols $l_1, l_2, \ldots$ belonging to some alphabet $\mathcal{L}$. Before observing the next symbol in the sequence, it predicts it by estimating the probability of observing each symbol of the alphabet. Then, it observes the symbol and some loss is incurred depending on the estimated probability of the current symbol. Subsequently, it adapts its model in order to better predict future symbols. The goal of *universal prediction* is to achieve an asymptotically optimal performance independently of the generating mechanism (see, e.g., the survey of Merhav and Feder [22]). When performance is measured in terms of the logarithmic loss, prediction is intimately related to data compression, gambling and investing (see, e.g., [7, 6]).

Barron's theorem [3] (see also [10, Ch.15]) establishes a fundamental link between prediction under logarithmic loss and learning: the better we can sequentially predict data from a probabilistic source, the faster we can identify a good approximation of it. This is of paramount importance when applied to nonparametric models of infinite dimensionality, where overfitting is a serious concern. This is our case, since the predictor observes some associated side information (i.e. *features*) $z_i \in \mathbb{R}^d$ before predicting $l_i \in \mathcal{L}$, where $\mathcal{L} = \{\lambda_1, \ldots, \lambda_{|\mathcal{L}|}\}$. We consider the probabilistic setup where the pairs of observations $(z_i, l_i)$ are i.i.d. realizations of some random variables $(Z, L)$ with joint probability measure $\mathbb{P}$. Therefore, we aim at estimating a nonparametric model of the conditional measure $\mathbb{P}_{L|Z}$.

Nonparametric distributions can be approximated by universal distributions over countable unions of parametric models (see e.g., [10, Ch. 13]). This approach requires defining parametric models that can arbitrarily approximate the nonparametric distribution as the number of parameters tend to infinity. For example, models based on histograms with arbitrarily many bins have been proposed to approximate univariate nonparametric densities (e.g., [13, 26, 36]).

Bayesian mixtures allow to obtain universal distributions for countable unions of parametric models (e.g., [35, 34]). Nevertheless, standard Bayesian mixtures suffer from the catch-up phenomenon, i.e., their convergence rate is not optimal. In [31], it has been shown that a better convergence rate can be achieved by allowing models to change over time, by considering, instead of a set of distributions $\mathcal{M}$, a (larger) set constituted by sequences of distributions of $\mathcal{M}$. The resulting *switch distribution* has still a Bayesian form but the mixture is done over sequences of models.

Previous works on prediction with side information either are non-sequential (e.g. PAC learning [30]), or use other losses (e.g. [11, 12] ) or consider side information in more restrictive spaces (e.g. [1, 5]). Our work bears similarities to [16, 29, 32] but the objectives are different and so are the guarantees. Recently, [19] proposed a universal online predictor for side information in $\mathbb{R}^d$ based on a mixture of nearest-neighbors regressors with different $k(n)$ functions specifying the number of neighbors at time $n$. Practically, the performances depend on the particular set of functions—a design choice—and its time complexity is linear in $n$ due to the exact nearest neighbor search. Gaussian Processes (see, e.g., [25]) are nonparametric Bayesian methods which can be used for online prediction with side information. It is conjectured that exact Gaussian processes with the radial basis function (RBF) kernel are universal under some conditions on the marginal measure $\mathbb{P}_Z$ [10, Sec. 13.5.3]. In practice, approximations are required to compute the predictive posterior for discrete labels (e.g. Laplace) and the kernel width strongly affects the results. In addition, their time complexity to predict each observation is $O\left(n^3\right)$, making them practical for small samples sizes only.

We propose a novel nonparametric online predictor with universal guarantees for continuous side information exhibiting two distinctive features. First, it relies on a hierarchical feature space discretization and a recursive Bayesian distribution, automatically learning the relevant feature scales and making it scale-hyperparameter free. Second, in contrast to other nonparametric approaches, its time complexity to process the $n$-th sample point is $O\left(\log n\right)$ in probability. Due to space constraints, proofs are presented in the supplementary material.

## 2 Basic definitions and notations

In order to represent sequences, we use the notation $x^n \equiv x_1, \ldots, x_n$. The functions $|\cdot|$ and $|\cdot|_\lambda$, give, respectively, the length of a sequence and the number of occurrences of a symbol $\lambda$ in it. Let $\mathbb{P}$ be the joint probability measure of $L, Z$. Let $\mathbb{P}_L, \mathbb{P}_Z$ be their respective marginal measures and $\mathbb{P}_{Z|L}$ the probability measure of $Z$ conditioned on $L$. The entropy of random variables is denoted $H\left(\cdot\right)$, while the entropy of $L$ conditioned on $Z$ is denoted $H\left(L|Z\right)$. The mutual information between $L$ and $Z$ is denoted $I\left(L; Z\right)$. Logarithms are taken in base 2.

A *finite-measurable partition* $A = (\gamma_1, \ldots, \gamma_n)$ of some set $\Omega$ is a subdivision of $\Omega$ into a finite number of disjoint measurable sets or *cells* $\gamma_i$ whose union is $\Omega$. An $n$-sample partition rule $\pi_n(\cdot)$ is a mapping from $\Omega^n$ to the space of finite-measurable partitions for $\Omega$, denoted $\mathcal{A}(\Omega)$. A partitioning scheme for $\Omega$ is a countable collection of $k$-sample partition rules $\Pi \equiv \{\pi_k\}_{k \in \mathbb{N}^+}$. The partitioning scheme at time $n$ defines the set of partition rules $\Pi_n \equiv \{\pi_k\}_{k=1..n}$. For a given $n$-sample partition rule $\pi_n(\cdot)$ and a sequence $z^n \in \Omega^n$, $\pi_n(z|z^n)$ denotes the unique cell in $\pi_n(z^n)$ containing a given point $z \in \Omega$. For a given partition $A$, let $A(z)$ denote the unique cell of $A$ containing $z$. Let $\gamma\left(\cdot\right)$ denote the operator that extracts the subsequences whose symbols have corresponding $z_i \in \gamma$.

## 3 The `kd-switch` distribution

We define the `kd-switch` distribution $P_{\text{kds}}$ using a k-d tree based hierarchical partitioning and a switch distribution defined over the union of multinomial distributions implied by the partitioning.

**Full-fledged k-d tree based spatial partitioning.** We obtain a hierarchical partitioning of $\Omega = \mathbb{R}^d$ using a full-fledged k-d tree [8, Sec. 20.4] that is naturally amenable to an online construction since pivot points are chosen in the same order as sample points are observed. Instead of rotating deterministically the axis of the projections, we sample the axis uniformly at each node of the tree. Formally, let $\Pi_{\text{kd}} \equiv \{\pi_k\}_{k \in \mathbb{N}^+}$ be the nested partitioning scheme such that $\pi_n(z^n)$ is the spatial partition generated by a full-fledged k-d tree after observing $z^n$. In order to define it recursively, let the base case be $\pi_0(z^0) \equiv \mathbb{R}^d$, where $z^0$ is the empty string. Then, $\pi_{n+1}(z^{n+1})$ is obtained by uniformly drawing a direction $J$ in $1..d$ and by replacing the cell $\gamma \in \pi_n(z^n)$ such that $z_{n+1} \in \gamma$ by

the following cells

$$\begin{cases} \gamma_1 \equiv \{z \in \gamma : z[J] \le z_{n+1}[J]\} \\ \gamma_2 \equiv \{z \in \gamma : z[J] > z_{n+1}[J]\} \end{cases} \tag{1}$$

where $\cdot[J]$ extracts the $J$-th coordinate of the given vector. A spatial partition $A = \{\gamma_1, \gamma_2, \ldots, \gamma_{|A|}\}$ of $\mathbb{R}^d$ defines a class of piecewise multinomial distributions characterized by $\boldsymbol{\theta}_A \equiv [\theta_1, \ldots, \theta_{|A|}]$, $\theta_i \in \Delta^{|\mathcal{L}|}$, where $\Delta^{|\mathcal{L}|}$ is the standard $|\mathcal{L}|$-simplex. More precisely, $P_{\boldsymbol{\theta}_A}(\cdot|z)$ is multinomial with parameter $\theta_i$ if $z \in \gamma_i$.

**Context Tree Switching.** We adapt the Context Tree Switching (CTS) distribution [33] to use spatial cells as contexts. Since these contexts are created as sample points $z_i$ are observed, the chronology of their creation has to be taken into account. Given a nested partitioning scheme $\Pi$ whose instantiation with $z^n$ creates a cell $\gamma$ and splits it into $\gamma_1$ and $\gamma_2$, we define the *cell splitting index* $\tau_n(\gamma)$ as the index in the subsequence $\gamma(z^n)$ when $\gamma_1$ and $\gamma_2$ are created (see Fig. 1). If $\gamma$ is not split by $\Pi$ instantiated with $z^n$, then we define $\tau_n(\gamma) \equiv \infty$.

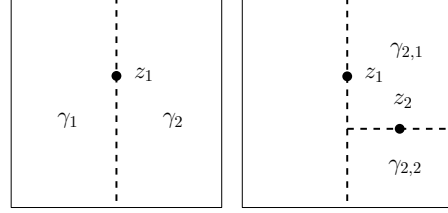

(a) $\pi_1(z^1)$: $\gamma_1$ and $\gamma_2$ are created, $z_1 \in \gamma_1$.

(b) $\pi_2(z^2)$: $\gamma_{2,1}$ and $\gamma_{2,2}$ are created. $z_2 \in \gamma_{2,2}$.

Figure 1: **Cell creation process and cell splitting index**. The cell splitting index is defined w.r.t. its subsequence: $\tau_2(\Omega) = 1, \tau_2(\gamma_2) = 1$ since $\gamma_2(z^2) = z_2$, and $\tau_2(\gamma_1) = \tau_2(\gamma_{2,1}) = \tau_2(\gamma_{2,2}) = \infty$.

At each cell $\gamma$, two models, defined later and denoted $a$ and $b$, are considered. Let $w_\gamma(\cdot)$ be a prior over model index sequences $i^m \equiv i_1, \ldots, i_m \in \{a, b\}^m$ at cell $\gamma$, recursively defined by

$$w_\gamma(i^m) \equiv \begin{cases} 1 \text{ if } m = 0 \\ \frac{1}{2} \text{ if } m = 1 \\ w_\gamma(i^{m-1})\left((1 - \alpha_m^\gamma)\mathbb{1}_E + \alpha_m^\gamma \mathbb{1}_{\neg E}\right) \text{ if } m > 1 \end{cases}, E \equiv \{i_m = i_{m-1}\}, \alpha_m^\gamma = m^{-1}.$$

In order to define the CTS distribution, we need the Jeffreys' mixture over multinomial distributions also known as the Krichevsky-Trofimov estimator [17]

$$P_{\mathrm{kt}}(l^n) \equiv \int_{\theta \in \Delta^{|\mathcal{L}|}} \prod_{j \in 1 \ldots |\mathcal{L}|} \theta[j]^{|l^n|_{\lambda_j}} w(\theta) d\theta \tag{2}$$

with $\theta[j]$ being the $j$-th component of the vector $\theta$, $|l^n|_{\lambda_j}$ the number of occurrences of $\lambda_j$ in $l^n$ and $w(\cdot)$ the Jeffreys' prior for the multinomial distribution [14] i.e. a Dirichlet distribution with parameters $(1/2, \ldots, 1/2)$.

Consider any cell $\gamma$ created by the partitioning scheme $\Pi$ instantiated with $z^n$. $\gamma$ can either be refined into two child cells $\gamma_1$ and $\gamma_2$ or have $\tau_n(\gamma) = \infty$. Given a sequence of labels $l^n$ such that all the corresponding positions $z_i \in \gamma$, the modified CTS distribution is given by

$$P_{\mathrm{cts}}^{\Pi,\gamma}(l^n|z^n) \equiv \sum_{i^n \in \{a,b\}^n} w_\gamma(i^n) \prod_{k=1}^n \left[\mathbb{1}_{\{i_k=a\}}\phi_a(l_k|l^{k-1}) + \mathbb{1}_{\{i_k=b\}}\phi_b^\gamma(l_k|l^{k-1}, z^k)\right] \tag{3}$$

where the predictive distributions of models $a$ and $b$ are given by

$$\phi_a(l_k|l^{k-1}) \equiv P_{\mathrm{kt}}(l_k|l^{k-1}) \equiv \frac{P_{\mathrm{kt}}(l^k)}{P_{\mathrm{kt}}(l^{k-1})} \tag{4}$$

and

$$\phi_b^\gamma(l_k|l^{k-1}, z^k) \equiv \begin{cases} P_{\mathrm{kt}}(l_k|l^{k-1}) \text{ if } k < \tau_k(\gamma) \\ \frac{P_{\mathrm{cts}}^{\Pi,\gamma_j}(\gamma_j(l^k)|\gamma_j(z^k))}{P_{\mathrm{cts}}^{\Pi,\gamma_j}(\gamma_j(l^k)^{-1}|\gamma_j(z^k)^{-1})} \text{ with } j : z_k \in \gamma_j, \text{ otherwise} \end{cases} \tag{5}$$

where $\cdot^{-1}$ removes the last symbol of a sequence and, for the empty sequences $l^0, z^0$, $P_{\mathrm{cts}}^{\Pi,\gamma}(l^0|z^0) \equiv 1$ and $P_{\mathrm{kt}}(l^0) \equiv 1$.

**Definition of $P_{\mathrm{kds}}$.** The `kd-switch` distribution is obtained from the modified CTS distribution on the context cells defined by the full-fledged k-d tree spatial partitioning scheme i.e.

$$P_{\mathrm{kds}}(l^n|z^n) \equiv P_{\mathrm{cts}}^{\Pi_{\mathrm{kd}},\mathbb{R}^d}(l^n|z^n). \tag{6}$$

**Remark 1.** *In [33], the authors observe better empirical performance with $\alpha_m^\gamma = n^{-1}$ for any cell $\gamma$, where $n$ is the number of samples observed at the root partition $\Omega$ when the $m$-th sample is observed in $\gamma$. With this switching rate they were able to provide a good redundancy bound for bounded depth trees. In our unbounded case, we observed a better empirical performance with $\alpha_m^\gamma = m^{-1}$.*

**Remark 2.** *A Context Tree Weighting [35] scheme can be obtained by setting $\alpha_m^\gamma = 0$. The corresponding distribution is denoted $P_{\mathrm{kdw}}$.*

## 4 Pointwise universality

In this section, we show that $P_{\mathrm{kds}}$ is pointwise universal, i.e. it achieves a normalized log loss asymptotically as good as the true conditional entropy of the source generating the samples. More formally, we state the following theorem.

**Theorem. 1.** *The `kd-switch` distribution is pointwise universal, i.e.*

$$-\lim_{n\to\infty} \frac{1}{n}\log P_{\mathrm{kds}}(L^n|Z^n) \leq H(L|Z) \ a.s. \tag{7}$$

*for any probability measure $\mathbb{P}$ generating the samples such that $\mathbb{P}_{Z|L}$ are absolutely continuous with respect to the Lebesgue measure.*

In order to prove Thm. 1, we first show that $P_{\mathrm{cts}}^{\Pi,\Omega}$ is universal with respect to the class of piecewise multinomial distributions defined by any nested partitioning scheme $\Pi$. Then, we show that $\Pi_{\mathrm{kd}}$ allows to approximate arbitrarily well any conditional distribution.

**Universality with respect to the class of piecewise multinomial distributions.** Consider a nested partitioning scheme $\Pi$ for $\Omega$. $\Pi_n$ instantiated with some $z^n \in \Omega^n$ naturally defines a tree structure whose root node represents $\Omega$. Given an arbitrary set of internal nodes, we can prune the tree by transforming these internal nodes into leaves and discarding the corresponding subtrees. The new set of leaf nodes define a partition of $\Omega$. Let $\mathcal{P}_n(z^n)$ be the set of all the partitions that can be obtained by pruning the tree induced by $\Pi_n$ instantiated with $z^n$.

The next lemma shows that $P_{\mathrm{cts}}^{\Pi,\Omega}$, defined in Eq. 3, is universal with respect to the class of piecewise multinomial distributions defined on the partitions $\mathcal{P}_n(z^n)$.

**Lemma. 1.** *Consider arbitrary sequences $l^n \in \mathcal{L}^n, z^n \in \Omega^n, n \geq 0$. Then, for any $A \in \mathcal{P}_n(z^n)$ and for any piecewise multinomial distribution $P_{\boldsymbol{\theta}_A}$, the following holds*

$$-\log P_{\mathrm{cts}}^{\Pi,\Omega}(l^n|z^n) \leq -\log P_{\boldsymbol{\theta}_A}(l^n|z^n) + |A|\zeta\left(\frac{n}{|A|}\right) + \Gamma_A \log 2n + O(1) \tag{8}$$

*and*

$$-\lim_{n\to\infty} \frac{1}{n}\log P_{\mathrm{cts}}^{\Pi,\Omega}(L^n|Z^n) \leq H(L|A(Z)) \ a.s. \tag{9}$$

*where $\Gamma_A$ is the number of nodes in the tree, induced by $\Pi_n$ instantiated with $z^n$, that represents $A$ (i.e., the code length given by a* natural *code for unbounded trees) and*

$$\zeta(x) \equiv \begin{cases} x\log|\mathcal{L}| & \text{if } 0 \leq x < 1 \\ \frac{|\mathcal{L}|-1}{2}\log x + \log|\mathcal{L}| & \text{if } x \geq 1 \end{cases}. \tag{10}$$

**Remark 3.** *In a Context Tree Weighting scheme ($\alpha_m^\gamma = 0$), the $\log 2n$ factor in Eq. 8 disappears. See proof of Lemma 1. Thus, universality holds for this case too.*

**Universal discretization of the feature space.** In order to prove that the k-d tree based partitions allow to approximate arbitrarily well the conditional entropy $H(L|Z)$, we use the following corollary of [28, Thm. 4.2].

**Corollary. 1.** *Let $\varnothing(\gamma) \equiv \sup_{x,y \in \gamma} \|x - y\|$. Let $\mathbb{P}$ be any probability measure such that $\mathbb{P}_{Z|L}$ are absolutely continuous with respect to the Lebesgue measure. Given a partition scheme $\Pi \equiv \{\pi_k\}_{k \in \mathbb{N}^+}$, if $\forall \delta > 0$*

$$\mathbb{P}_Z \left( \{ z \in \mathbb{R}^d : \varnothing(\pi_n(z|Z^n)) > \delta \} \right) \xrightarrow{a.s.} 0 \tag{11}$$

*then $\Pi$ universally discretizes the feature space, i.e.*

$$H\left(L | \pi_n(Z|Z^n)\right) \xrightarrow{a.s.} H\left(L|Z\right). \tag{12}$$

The next lemma provides the required shrinking condition for the k-d tree based partitioning.

**Lemma. 2.** *$\Pi_{kd}$ satisfies the shrinking condition of Eq. 11 and, thus, universally discretizes the feature space.*

**Pointwise universality.** The proof of Thm. 1 on the pointwise universality of $P_{\text{kds}}$ stems from a combination of Lemmas 1 and 2—see Appendix.

# 5 Online algorithm

Since a direct computation of Eq. (3) is intractable and an online implementation is desired, we use the recursive form of [33, Algorithm 1], which performs the exact calculation. We denote by $P_{\text{s}}^\gamma$ the sequentially computed `kd-switch` distribution at node $\gamma$. In Section 5.2, we show that $P_{\text{s}}^\gamma(l^n|z^n) = P_{\text{cts}}^{\Pi_{kd},\gamma}(l^n|z^n)$.

## 5.1 Algorithm

**Outline.** For each node of the k-d tree, the algorithm maintains two weights denoted $w_\gamma^a$ and $w_\gamma^b$. As follows from [33, Lemma 2], if $l^t, z^t$ are the subsequences observed in $\gamma$ and $w_\gamma^a$ is the weight before processing $l_t$, then, $w_\gamma^a P_{\text{kt}}\left(l_t|l^{t-1}\right)$ corresponds to the contribution of all possible model sequences ending in model $a$ (KT) to the total probability assigned to $l^t$ by the CTS distribution. Analogously, $w_\gamma^b P_{\text{s}}^\gamma\left(l_t|l^{t-1}, z^t\right)$ corresponds to the contribution of all possible model sequences ending in model $b$ (CTS).

We now describe the three steps that allow the online computation of $P_{\text{s}}^\gamma(l^n|z^n)$ given by Eq. 14. The algorithm starts with only a root node representing $\mathbb{R}^d$. When a new point $z_*$ is observed, the following steps are performed.

**Step 1: k-d tree update and new cells' initialization.** The point $z_*$ is passed down the k-d tree until it reaches a leaf cell $\gamma$. Then, a coordinate $J$ is uniformly drawn from $1 \ldots d$ and two child nodes, corresponding to the new cells defined in Eq. 1 with $z_*$ as splitting point, are created.

Let $l^n, z^n$ be the subsequences observed in $\gamma$ and thus $z_n = z_*$. Since the new cells may contain some of the symbols in $l^{n-1}$, the following initialization is performed at each new node $\gamma_i, i \in \{1, 2\}$:

$$\begin{aligned} w_{\gamma_i}^a &\leftarrow \frac{1}{2} P_{\text{kt}}\left(\gamma_i\left(l^{n-1}\right)\right) \\ w_{\gamma_i}^b &\leftarrow \frac{1}{2} P_{\text{kt}}\left(\gamma_i\left(l^{n-1}\right)\right) \end{aligned}, \text{ with } P_{\text{kt}}\left(\gamma_i\left(l^{n-1}\right)\right) = 1 \text{ if } \gamma_i\left(l^{n-1}\right) \text{ is empty.} \tag{13}$$

**Step 2: Prediction.** The probability assigned to the subsequence $l^n$ given $z^n$ observed in $\gamma$ is

$$P_{\text{s}}^\gamma(l^n|z^n) \leftarrow w_\gamma^a P_{\text{kt}}\left(l_n|l^{n-1}\right) + w_\gamma^b P_{\text{r}}^\gamma\left(l_n|l^{n-1}, z^n\right) \tag{14}$$

where

$$P_{\text{r}}^\gamma\left(l_n|l^{n-1}, z^n\right) \leftarrow \begin{cases} P_{\text{kt}}\left(l_n|l^{n-1}\right) \text{ if } n < \tau_n(\gamma) \\ \frac{P_{\text{s}}^{\gamma_j}(\gamma_j(l^n)|\gamma_j(z^n))}{P_{\text{s}}^{\gamma_j}(\gamma_j(l^n)^{-1}|\gamma_j(z^n)^{-1})} \text{ with } j : z_n \in \gamma_j, \text{ otherwise} \end{cases}. \tag{15}$$

**Step 3: Updates.** Having computed the probability assignment of Eq. 14, the weights of the nodes corresponding to the cells $\{\gamma : z_* \in \gamma\}$ are updated. Given a node $\gamma$ to be updated, let $l^n, z^n$ be the subsequences observed in $\gamma$. The following updates are applied:

$$\begin{aligned} w_\gamma^a &\leftarrow \alpha_{n+1}^\gamma P_{\text{s}}^\gamma(l^n|z^n) + \beta_{n+1}^\gamma w_\gamma^a P_{\text{kt}}\left(l_n|l^{n-1}\right) \\ w_\gamma^b &\leftarrow \alpha_{n+1}^\gamma P_{\text{s}}^\gamma(l^n|z^n) + \beta_{n+1}^\gamma w_\gamma^b P_{\text{r}}^\gamma\left(l_n|l^{n-1}, z^n\right) \end{aligned} \tag{16}$$

where $\beta_n^\gamma \equiv (1 - 2\alpha_n^\gamma)$. When $\gamma$ has just been created (i.e. $\gamma$ is a leaf node), these updates reduce to

$$
\begin{aligned}
w_\gamma^a &\leftarrow w_\gamma^a P_{\mathrm{kt}}\big(l_n|l^{n-1}\big) \\
w_\gamma^b &\leftarrow w_\gamma^b P_{\mathrm{kt}}\big(l_n|l^{n-1}\big)
\end{aligned}
\tag{17}
$$

**Remark 4.** *The KT estimator can be computed sequentially using the following formula [27]:*

$$
P_{\mathrm{kt}}\big(l_n|l^{n-1}\big) = \frac{\big|l^{n-1}\big|_{l_n} + \frac{1}{2}}{\big|l^{n-1}\big| + \frac{|\mathcal{L}|}{2}}.
\tag{18}
$$

*Therefore, the sequential computation only requires maintaining the counters $\big|l^{n-1}\big|_{l_n}$ for each cell.*

**Remark 5.** *Samples $z_i$ only need to be stored at leaf nodes. Once a leaf node is split, they are moved to their corresponding child nodes.*

## 5.2 Correctness

The steps of our algorithm are the same as those of [33, Algorithm 1] except for the initialization of Eq. 13. In fact, as shown in the next lemma, it is equivalent to building the partitioning tree from the beginning (assuming, without loss of generality, that $z^n$ is known in advance) and applying the original algorithm at every relevant context.

**Lemma. 3.** *Let $n \in \mathbb{N}^+$ and assume the partitioning tree for $z^n$ is built from the beginning. Let $\gamma$ be any node of the tree. If the original initialization and update equations from [33, Algorithm 1] (corresponding respectively to Eq. 13 with an empty sequence and Eq. 16) are applied, the weights, after observing $l^t$ in $\gamma$ with $t < \tau_n(\gamma)$, are $w_\gamma^a = \frac{1}{2} P_{\mathrm{kt}}(l^t)$ and $w_\gamma^b = \frac{1}{2} P_{\mathrm{kt}}(l^t)$, which correspond to those obtained after the initialization of Eq. 13 and the updates of Eq. 17.*

The correctness of our algorithm follows from Lem. 3 and [33, Thm. 4], since for $t \geq \tau_n(\gamma)$ the original update equations are used.

## 5.3 Complexity

The cost of processing $l_n, z_n$ is linear in the depth $D_n$ of the node split by the insertion of $z_n$, since the algorithm updates the weights at each node in the path leading to this node. If $\mathbb{P}_Z$ is absolutely continuous with respect to the Lebesgue measure, since the full-fledged k-d tree is monotone transformation invariant, we can assume without loss of generality that the marginal distributions of $Z$ are uniform in $[0, 1]$ (see [8, Sec. 20.1]) and thus its profile is equivalent to that of a random binary search tree under the random permutation model (see [21, Sec. 2.3]). Then, $D_n$ corresponds to the cost of an unsuccessful search and $\frac{D_n}{2\log n} \to 1$ in probability (see [21, Sec. 2.4]). Therefore, the complexity of processing $l_n, z_n$ is $O(\log n)$ in probability with respect to $\mathbb{P}_{Z^n}$.

# 6 Experiments

**Software-hardware setup.** Python code and data used for the experiments are available at `https://github.com/alherit/kd-switch`. Experiments were carried out on a machine running Debian 3.16, equipped with two Intel(R) Xeon(R) E5-2667 v2 @ 3.30GHz processors and 62 GB of RAM.

**Boosting finite length performance with ensembling.** When considering finite length performance, we can be unlucky and obtain a bad set of hierarchical partitions (i.e., with low discrimination power). In order to boost the probability of finding good partitions, we can use a Bayesian mixture of J trees. Bayesian mixtures trivially maintain universality.

**Two sampling scenarii for labels.** In the first one, labels are sampled from a Bernoulli distribution such that $\mathbb{P}(L = 0) = \theta_0$, where $\theta_0$ is a known parameter. We then we sample from $\mathbb{P}_{Z|L}$. In this case, the root node distribution $P_{\mathrm{kt}}\big(l_n|l^{n-1}\big)$ is replaced by $\mathbb{P}(L_n = l_n) = \theta_0^{\mathbb{1}_{\{l_n=0\}}}(1 - \theta_0)^{\mathbb{1}_{\{l_n=1\}}}$, since $\theta_0$ is known. In the second one, observations come in random order and $\mathbb{P}(L)$ is unknown.

## 6.1 Normalized log loss (NLL) convergence

**Datasets.** We use the following datasets, detailed in Appendix B.1: **(L-i)** A $2D$ dataset consists of two Gaussian Mixtures spanning three different scales. **(L-ii)** A dataset in dimension $d = 784$

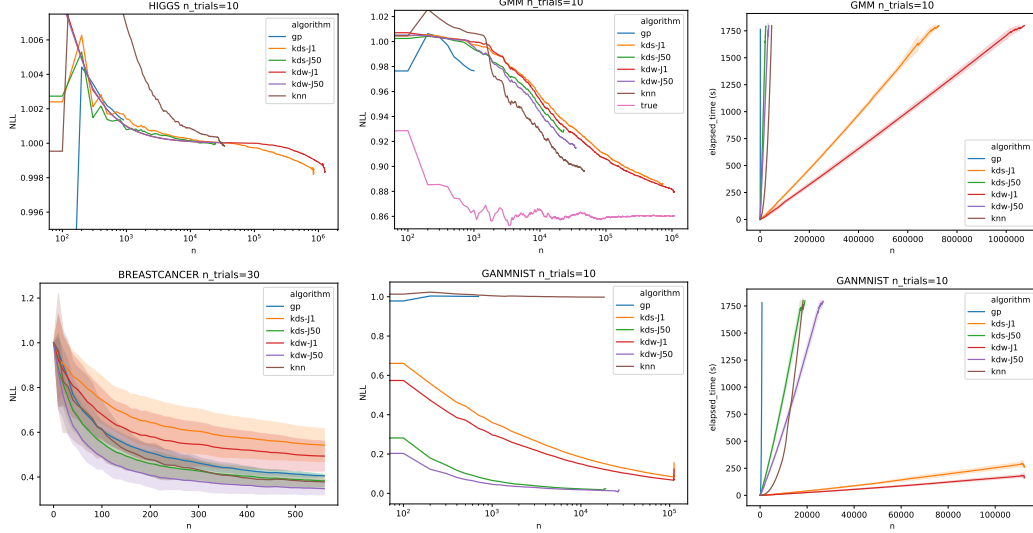

Figure 2: **(Left and Middle)** Convergence of NLL as a function of $n$, for a 30' calculation. **(Right)** Running time as a function of $n$. Error bands represent the std dev. w.r.t. the randomness in the tree generation except for dataset (L-iv) where they represent the std dev. w.r.t. the shuffling of the data.

composed of both real MNIST digits, as well as digits generated by a Generative Adversarial Network [24] trained on the MNIST dataset. **(L-iii)** The Higgs dataset [20], the goal being to distinguish the signature of processes producing Higgs bosons. **(L-iv)** The Breast Cancer Wisconsin (Diagnostic) Data Set [20]—dimension $d = 30$.

For cases (L-i,L-ii,L-iii), in order to feed the online predictors, we apply the first sampling scenario for labels. For case (L-iv), we apply the second one and, in each trial, we take the pooled dataset in a random order to feed the online predictors.

**Results.** We focus on the cumulative normalized log loss performance (NLL), and the trade-off with the computational requirements—by limiting the running time to 30'.

We compare the performance of our online predictors $P_{kds}$ and $P_{kdw}$ (see Rmk. 2) with a number of trees $J \in \{1, 50\}$, against the following contenders. The Bayesian mixture of knn-based sequential regressors proposed in [19], with a switch distribution using a horizon-free prior as $P_{kds}$. Practically, this predictor depends on a given set of functions of $n$ specifying the number of neighbors. We use the same set specified in [19]. We also consider a Bayesian Mixture of Gaussian Processes Classifiers (gp) with RBF kernel width $\sigma \in \{2^{4i}\}_{i=-5...7}$. (Our implementation uses the scikit-learn GaussianProcessClassifier [23]. For each observation, we retrain the classifier using all past observations—a step requiring samples from the two populations. Thus, we predict with a uniform distribution (or $\mathbb{P}(L)$ when known) until at least one instance of each label has been observed.) In the case (L-i), we also compare to the true conditional probability, which can be easily derived since $\mathbb{P}_{Z|L}$ are known. Note that the normalized log loss of the true conditional probability converges to the conditional entropy by the Shannon–McMillan–Breiman theorem [7].

Fig. 2 (Left and Middle) shows the NLL convergence with respect to the number of samples. Notice that due to the 30' running time budget, curves stop at different $n$. Fig. 2 (Right) illustrates the computational complexity of each method. For our predictors, the statistical efficiency increases with the number of trees—at the expense of the computational burden. Weighting performs better than switching for datasets (L-i, L-ii, L-iv), and the other way around for (L-iii). knn takes some time to get the *right scale*, then converges fast in most cases—with a plateau though on dataset L-ii though. This owes to the particular set of exponents used to define the mixture of regressors [19]. Also, knn is computationally demanding, in particular when compared to our predictors with $J = 1$.

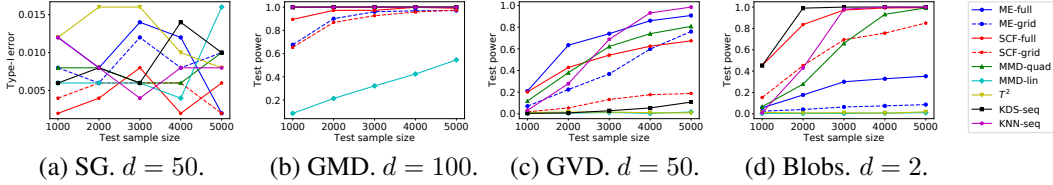

| | |
|---|---|
| (a) SG. $d = 50$. | (b) GMD. $d = 100$. |
| (c) GVD. $d = 50$. | (d) Blobs. $d = 2$. |

Figure 3: **Tests on randomly rotated Gaussian datasets from [15].** The abscissa represents the test sample size $n_{\text{test}}$ for each of the two samples. Thus, for sequential methods, $n = 4n_{\text{test}}$.

### 6.2 Two-sample testing (TST)

**Construction.** Given samples from two distributions, whose corresponding random variables $X \in \mathbb{R}^d$ and $Y \in \mathbb{R}^d$ are i.i.d., a nonparametric two-sample test tries to determine whether the null hypothesis $\mathbb{P}_X = \mathbb{P}_Y$ holds or not (see, e.g., [18, Section 6.9]). Consistent sequential two-sample tests with optional stop (i.e. the $p$-value is valid at any time $n$) can be built from a pointwise universal online predictor $Q$ [19] by defining $(L, Z)$ as: $(0, X)$ with probability $\theta_0$, or $(1, Y)$ with probability $1 - \theta_0$, where $\theta_0$ is a design parameter set to $1/2$ in the following experiments. The $p$-value is the likelihood ratio $\frac{\mathbb{P}(l^n)}{Q(l^n|z^n)}$. Note this corresponds to the first sampling scenario for labels. The instantiation of this construction with $P_{\text{kds}}$ and $J = 50$ is denoted `KDS-seq`.

**Contenders.** We compare `KDS-seq` against $SW_{\pi_S}$ from [19], denoted `KNN-seq`: a sequential two-sample test obtained by instantiating the construction described above with the online knn predictor described in the previous section. We also compare `KDS-seq` against the kernel tests from [15]: `ME-full`, `ME-grid`, `SCF-full`, `SCF-grid`, `MMD-quad`, `MMD-lin`, and the classical Hotelling's $T^2$ for differences in means under Gaussian assumptions. These tests depend on a kernel width $\sigma$ learned on a trained set—the train-test paradigm—as opposed to `KDS-seq` which automatically detects the pertinent scales. Contenders were launched with the hyperparameters specified in their respective paper. For a fair comparison between sequential methods and those tests using the train-test paradigm with $n_{\text{test}}$ used for testing, we use a number of samples $n = 4n_{\text{test}}$—detail in Appendix B.2.

**Datasets.** We use the four datasets from [15, Table 1]: **(T-i)** Same Gaussians in dimension $d = 50$, to assess the type I error; **(T-ii)** Gaussian Mean Difference (GMD): normal distributions with a difference in means along one direction, $d = 100$; **(T-iii)** Gaussian Variance Difference (GVD): normal distributions with a difference in variance along one direction, $d = 50$; **(T-iv)** Blobs (Mixture of Gaussian distributions centered on a lattice) [9]. Datasets (T-ii, T-iii, T-iv) are meant to assess type II error. To prevent k-d tree cuts to exploit the particular direction where the difference lies, these datasets undergo a random rotation (one per tree). See Appendix B.3 for results without rotations.

**Results.** The significance level is set to $\alpha = .01$ in all the cases. The Type I error rate and the power $(1 - \text{Type II error rate})$ are computed over 500 trials. In the SG case (Fig. 3(a)), all the tests have a Type I error rate around the specified $\alpha$ as expected. In the GMD and Blobs cases (Fig. 3(b,d)), `KDS-seq` matches or outperforms all the contenders. On Blobs, `KDS-seq` outperforms `KNN-seq` thanks to its automatic scale detection, even though the mixture used by the latter allows it to handle the multiple scales. For GVD (Fig. 3(c)), our results are weaker. To see why, recall that GMD is generated by adding one unit to one coordinate of the mean vector, while GVD is obtained by doubling the variance along one direction. The span of the latter dataset is larger, and upon rotating the data—see comment above—all directions are impacted. Given the high dimensionality, the partitioning of k-d trees faces more difficulties to reduce the diameter of cells, which is key to convergence—see Corollary 1.

## 7 Outlook

We foresee the following research directions. A first open question is to characterize the situations where switching should be preferred over weighting. A second core question is to quantify the ability of our k-d tree based construction to cope with multiple scales in the data. A third one is the derivation of finite length bounds related to the *complexity* of the underlying conditional distribution. Finally, accommodating data in a metric (non Euclidean) space, using e.g. metric trees, would widen the application spectrum of the method.

**Acknowledgments**

We would like to thank María Zuluaga, Eoin Thomas, Nicolas Bondoux and Rodrigo Acuña-Agost for insightful comments and, also, Wittawat Jitkrittum and Arthur Gretton for providing us the complete output of their experiments.

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
