[Supplementary Material]

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

# A  Proofs

*Lemma 1.* We follow the lines of the proof of [33, Thm. 3], except that we need to take into account the chronology of cell creation. For notational convenience we omit $\Pi$ in $P_{\text{cts}}^{\Pi,\cdot}$.

Let $\tilde{A}$ denote the set of cells corresponding to internal nodes in the tree structure leading to $A$. Given a cell $\gamma$ and $l^n, z^n$, let us denote $n_\gamma \equiv |\gamma(z^n)|$, $\underline{l}^{n_\gamma} \equiv \gamma(l^n)$ and $\underline{z}^{n_\gamma} \equiv \gamma(z^n)$. Then, $\forall n \geq 1, \forall l^n \in \mathcal{L}^n, \forall z^n \in \Omega^n$, by dropping the sum in Equation 3 and choosing the model index sequences $bb\ldots b$ for internal nodes and $aa\ldots a$ for leaf nodes, we have that for any cell $\gamma \in \tilde{A}$

$$
\begin{aligned}
P_{\text{cts}}^\gamma(\gamma(l^n)|\gamma(z^n)) &\geq w_\gamma(b^{n_\gamma}) P_{\text{kt}}\left(\underline{l}^{\tau_n(\gamma)-1}\right) \prod_{k=\tau_n(\gamma)}^{n_\gamma} \sum_{j=1}^{2} \mathbb{1}_{\{\underline{z}_k \in \gamma_j\}} \frac{P_{\text{cts}}^{\gamma_j}\left(\gamma_j\left(\underline{l}^k\right)|\gamma_j\left(\underline{z}^k\right)\right)}{P_{\text{cts}}^{\gamma_j}\left(\gamma_j\left(\underline{l}^k\right)^{-1}|\gamma_j\left(\underline{z}^k\right)^{-1}\right)} \\
&= w_\gamma(b^{n_\gamma}) P_{\text{kt}}\left(\underline{l}^{\tau_n(\gamma)-1}\right) \prod_{j=1}^{2} \frac{P_{\text{cts}}^{\gamma_j}(\gamma_j\left(\underline{l}^{n_\gamma}\right)|\gamma_j\left(\underline{z}^{n_\gamma}\right))}{P_{\text{cts}}^{\gamma_j}\left(\gamma_j\left(\underline{l}^{\tau_n(\gamma)}\right)^{-1}|\gamma_j\left(\underline{z}^{\tau_n(\gamma)}\right)^{-1}\right)}
\end{aligned}
\tag{19}
$$

and for any cell $\gamma \in A$

$$
P_{\text{cts}}^\gamma(\gamma(l^n)|\gamma(z^n)) \geq w_\gamma(a^{n_\gamma}) P_{\text{kt}}(\gamma(l^n)) .
\tag{20}
$$

Then, by repeatedly applying Eq. 19 at internal nodes and Eq. 20 at leaf nodes, we obtain

$$
P_{\text{cts}}^\Omega(l^n|z^n) \geq \left(\prod_{\gamma \in A} w_\gamma(a^{n_\gamma})\right) \left(\prod_{\gamma \in \tilde{A}} w_\gamma(b^{n_\gamma})\right) \left(\prod_{\gamma \in A} P_{\text{kt}}(\gamma(l^n))\right) \kappa
\tag{21}
$$

where $\kappa$ groups all the terms from Eq. 19 that do not depend on $n$. We have

$$
w_\gamma(a^{n_\gamma}) = \frac{1}{2} \prod_{t=2}^{n_\gamma} \frac{t-1}{t} = \frac{1}{2n_\gamma} \geq \frac{1}{2n}
\tag{22}
$$

where the factor $\frac{t-1}{t}$ for $t \geq 2$ comes from the prior probability of not switching. Analogously, $w_\gamma(b^{n_\gamma}) \geq \frac{1}{2n}$. (Note that, in a Context Tree Weighting scheme (see Remark 2), the prior probability of not switching is 1 and thus, the lower bound for $w_\gamma(a^{n_\gamma})$ and $w_\gamma(b^{n_\gamma})$ becomes $1/2$.) Then,

$$
P_{\text{cts}}^\Omega(l^n|z^n) \geq \kappa(2n)^{-\Gamma_A} \prod_{\gamma \in A} P_{\text{kt}}(\gamma(l^n)) .
\tag{23}
$$

The claimed inequality follows since, from [35, Eq. 23] (see e.g. [4, Eq. 17] for $|\mathcal{L}| > 2$),

$$
-\log \prod_{\gamma \in A} P_{\text{kt}}(\gamma(l^n)) \leq |A|\zeta\left(\frac{n}{|A|}\right) - \log P_{\boldsymbol{\theta}_A}(l^n|z^n) .
\tag{24}
$$

The limit follows from the Shannon–McMillan–Breiman theorem (see, e.g., [7]). $\qquad\square$

*Corollary 1.* The conditional entropy can be written as

$$
H(L|Z) = H(L) - I(L;Z)
\tag{25}
$$
$$
= H(L) - \mathbb{E}_{\mathbb{P}_L}\left[D_{\text{KL}}\left(\mathbb{P}_{Z|L}\|\mathbb{P}_Z\right)\right]
\tag{26}
$$

where $D_{\text{KL}}(\cdot\|\cdot)$ denotes the Kullback-Leibler divergence. Since $\mathbb{P}_{Z|L}$ are absolutely continuous with respect to $\mathbb{P}_Z$ and these measures are absolutely continuous with respect to the Lebesgue measure, the claim follows from [28, Thm. 4.2], which guarantees

$$
D_{\text{KL}}\left(\mathbb{P}_{\pi_n(Z|Z^n)|L}\|\mathbb{P}_{\pi_n(Z|Z^n)}\right) \xrightarrow{\text{a.s.}} D_{\text{KL}}\left(\mathbb{P}_{Z|L}\|\mathbb{P}_Z\right) . \qquad\square
$$

*Lemma 2.* By Markov's inequality, it is sufficient to show that

$$
\mathbb{E}_{\mathbb{P}_Z}\left[\varnothing(\pi_n(Z|Z^n))\right] \xrightarrow{n\to\infty} 0 \text{ a.s..}
\tag{27}
$$

As in [8, Sec. 20.1], since the k-d tree is monotone transformation invariant, we can assume without loss of generality that $Z \in [0,1]^d$. In the proof of [8, Thm. 20.3], it is shown that for any $\epsilon > 0$ and any $x \in \mathbb{R}^d$

$$\varnothing(\pi_n(x|Z^n)) \leq 2\epsilon\sqrt{d} \tag{28}$$

if some specific event $E(x, Z^n, \epsilon)$ holds. (Devroye presents the proof for the case $d = 2$, leaving the straightforward adaptation for $d > 2$ to the reader. The proof considers a fixed point $x$ and, upon inserting $k$ points into the k-d tree, the maximum distance from $x$ to the $2^d$ faces of its containing hyper-rectangle. The event $E(x, Z^n, \epsilon)$ stipulates that this maximum distance is bounded by $\epsilon$ upon inserting a number of points which may be $k = n^{1/3}$ or $k = n^{2/3}$ or $k = n$.) Then, it is shown that one can construct a set $B \subset \mathbb{R}^d$ such that $\mathbb{P}_Z(Z \in B) = 1$ and for all $x \in B$, and sufficiently small $\epsilon > 0$, $\mathbb{P}_{Z^n}(E(x, Z^n, \epsilon)) \xrightarrow{n\to\infty} 1$.

Then, for $\epsilon > 0$ sufficiently small, by total probability, we have

$$\mathbb{P}_{Z^n,Z}(E(Z, Z^n, \epsilon)) \tag{29}$$

$$= \mathbb{P}_{Z^n,Z}(E(Z, Z^n, \epsilon)|Z \in B)\,\mathbb{P}_Z(Z \in B) + \mathbb{P}_{Z^n,Z}(E(Z, Z^n, \epsilon)|Z \notin B)\,\mathbb{P}_Z(Z \notin B) \tag{30}$$

$$= \mathbb{P}_{Z^n,Z}(E(Z, Z^n, \epsilon)|Z \in B). \tag{31}$$

Since the desired property is defined w.r.t. $\mathbb{P}_Z$ while Eq. (28) involves the sequence $Z^n$, we apply the law of total expectation with the two events $E(Z, Z^n, \epsilon)$ and $\neg E(Z, Z^n, \epsilon)$:

$$\lim_{n\to\infty} \mathbb{E}_{\mathbb{P}_Z}[\varnothing(\pi_n(Z|Z^n))] =$$

$$\lim_{n\to\infty} \mathbb{E}_{\mathbb{P}_Z}[\varnothing(\pi_n(Z|Z^n))|E(Z, Z^n, \epsilon)]\,\mathbb{P}_{Z^n,Z}(E(Z, Z^n, \epsilon)|Z \in B) + \tag{32}$$

$$\mathbb{E}_{\mathbb{P}_Z}[\varnothing(\pi_n(Z|Z^n))|\neg E(Z, Z^n, \epsilon)]\,\mathbb{P}_{Z^n,Z}(\neg E(Z, Z^n, \epsilon)|Z \in B) =$$

$$\lim_{n\to\infty} \mathbb{E}_{\mathbb{P}_Z}[\varnothing(\pi_n(Z|Z^n))|E(Z, Z^n, \epsilon)] \cdot \lim_{n\to\infty} \mathbb{P}_{Z^n,Z}(E(Z, Z^n, \epsilon)|Z \in B) +$$

$$\lim_{n\to\infty} \mathbb{E}_{\mathbb{P}_Z}[\varnothing(\pi_n(Z|Z^n))|\neg E(Z, Z^n, \epsilon)] \cdot \lim_{n\to\infty} \mathbb{P}_{Z^n,Z}(\neg E(Z, Z^n, \epsilon)|Z \in B) = \tag{33}$$

$$\lim_{n\to\infty} \mathbb{E}_{\mathbb{P}_Z}[\varnothing(\pi_n(Z|Z^n))|E(Z, Z^n, \epsilon)] \tag{34}$$

where the last equality stems from $\varnothing(\pi_n(Z|Z^n)) < \infty$ and $\lim_{n\to\infty} \mathbb{P}_{Z^n,Z}(E(Z, Z^n, \epsilon)|Z \in B) = 1$, for sufficiently small $\epsilon > 0$. The random variable $\varnothing(\pi_n(Z|Z^n)$ is bounded since $Z \in [0,1]^d$. Therefore, by Lebesgue's dominated convergence theorem:

$$\lim_{n\to\infty} \mathbb{E}_{\mathbb{P}_Z}[\varnothing(\pi_n(Z|Z^n))|E(Z, Z^n, \epsilon)] = \mathbb{E}_{\mathbb{P}_Z}\left[\lim_{n\to\infty} \varnothing(\pi_n(Z|Z^n))|E(Z, Z^n, \epsilon)\right] = 0 \tag{35}$$

since $\varnothing(\pi_n(Z|Z^n)) \leq 2\epsilon\sqrt{d}$ if $E(Z, Z^n, \epsilon)$ holds, for any $\epsilon > 0$. Therefore,

$$\mathbb{P}_{Z^n}\left(\lim_{n\to\infty} \mathbb{E}_{\mathbb{P}_Z}[\varnothing(\pi_n(Z|Z^n))] = 0\right) = 1. \qquad \square$$

*Proof of Theorem 1.* By Lemma 2, for any $\varepsilon > 0$, there exists $n(\varepsilon)$ such that $\forall n \geq n(\varepsilon)$

$$H(L|\pi_n(Z|Z^n)) < H(L|Z) + \varepsilon \text{ a.s..} \tag{36}$$

Then, by Lemma 1

$$-\lim_{n\to\infty} \frac{1}{n} \log P_{\text{kds}}(L^n|Z^n) \leq H\left(L|\pi_{n(\varepsilon)}\left(Z|Z^{n(\varepsilon)}\right)\right) \text{ a.s..} \tag{37}$$

The claim follows since $\varepsilon$ can be arbitrary small. $\qquad \square$

*Lemma 3.* We use induction on $t$ and we denote $w_{\gamma,t}^a$ the value of $w_\gamma^a$ at the end of the updates after observing $l^t$ in $\gamma$. For $t = 0$, it trivially holds since the observed sequence is empty. For $1 \leq t < \tau_n(\gamma)$, by Eq. 14 and 16

$$w_{\gamma,t}^a = \alpha_{t+1}^\gamma P_{\text{kt}}(l^t) + \beta_{t+1}^\gamma w_{\gamma,t-1}^a P_{\text{kt}}(l_t|l^{t-1})$$
$$w_{\gamma,t}^b = \alpha_{t+1}^\gamma P_{\text{kt}}(l^t) + \beta_{t+1}^\gamma w_{\gamma,t-1}^b P_{\text{kt}}(l_t|l^{t-1}). \tag{38}$$

Using the inductive hypothesis, we get

$$w_{\gamma,t}^a = \alpha_{t+1}^\gamma P_{\text{kt}}(l^t) + \beta_{t+1}^\gamma \frac{1}{2} P_{\text{kt}}(l^t) = \frac{1}{2} P_{\text{kt}}(l^t)$$
$$w_{\gamma,t}^b = \alpha_{t+1}^\gamma P_{\text{kt}}(l^t) + \beta_{t+1}^\gamma \frac{1}{2} P_{\text{kt}}(l^t) = \frac{1}{2} P_{\text{kt}}(l^t). \tag{39}$$

$$\square$$

Figure 4: **Multiscale Gaussian Mixture dataset.** Union of two sample sets, each from a random mixture model.

Figure 5: **(Left.)** Examples of real images. **(Right.)** Examples of GAN generated synthetic images.

# B   Experiments

## B.1   Datasets for the normalized log loss convergence analysis

**Multiscale Gaussian Mixture dataset.**   The Gaussian mixture dataset from Section 6.1 is built by generating two Gaussian Mixture models, one for each label. The means of the Gaussians are uniformly drawn from $[0, 1]^2$ and are the same for both mixtures. The weights are randomly drawn from a Dirichlet distribution with parameters $(1, \ldots, 1)$ and are the same for both the mixtures. The covariance matrices are randomly drawn from an inverse Wishart distribution with $d + 2$ degrees of freedom and a scale parameter $I$ for a third of the components, $.01I$ for the second third and $.0001I$ for the last one. See Fig. 4 for an illustration.

**GAN dataset.**   We used the pretrained Deep Convolutional Generative Adversarial Network available at `https://github.com/csinva/pytorch_gan_pretrained`. We generate as many samples as real ones (60000). We consider the real samples as coming from $P_{Z|L=0}$ and the synthetic ones from $P_{Z|L=1}$. See Fig. 5 for an illustration.

**HIGGS dataset.**   To distinguish the signature of processes producing Higgs bosons from background processes which do not, we use the four low-level features (azimuthal angular momenta $\phi$ for four particle jets) which are known to carry very little discriminating information [20, 2].

## B.2   Datasets and sampling for the Two-sample-test experiments

**Datasets.**   We use the four datasets from [15, Table 1]: Same Gaussian (SG; $\mathbb{P}_X$ and $\mathbb{P}_Y$ are identical normal distributions; $d = 50$); Gaussian Mean Difference (GMD; $\mathbb{P}_X$ and $\mathbb{P}_Y$ are normal distributions with a difference in means along one direction; $d = 100$); Gaussian Variance Difference (GVD; $\mathbb{P}_X$ and $\mathbb{P}_Y$ are normal distributions with a difference in variance along one direction; $d = 50$); Blobs

| | |
| --- | --- |
| (a) SG. $d = 50$. | (b) GMD. $d = 100$. | (c) GVD. $d = 50$. | (d) Blobs. $d = 2$. |

Figure 6: **Tests on non-rotated Gaussian datasets, as specified in [15].** The abscissa represents the test sample size $n_{\text{test}}$ for each of the two samples. Thus, for sequential methods, $n = 4n_{\text{test}}$.

(Mixture of Gaussian distributions centered on a lattice)—a challenging case since differences occur at a much smaller length scale compared to the global scale [9]. To prevent k-d tree cuts to exploit the particular direction where the difference lies, such datasets undergo a random rotation (one per tree).

**Sampling.** For a fair comparison against tests using the train-test paradigm, sequential two-sample tests use a sample size equal to the sum of the training and test set sizes used by the contenders. When we compare to these tests, samples are obtained by the same sampling mechanism and with the same random seed, using the code provided at `https://github.com/wittawatj/interpretable-test`. Sequential tests (i.e. `KDS-seq` and `KNN-seq`) consume these samples following the first sampling scenario specified in Section 6, with $\theta_0 = .5$—labels are balanced.

## B.3 Two-sample-test experiments without random rotations

Figure 6 shows the results on the original datasets without undergoing random rotations. We observe that k-d tree cuts are able to quickly detect the particular directions where the difference lies making the power significantly higher than for the randomly rotated case.