[Reviews · NeurIPS 2019]

Reviewer 1



The paper proposed a novel distribution, the kd-switch distribution using a k-d tree based hierarchical partitioning and a switch distribution as a nonparametric prior distribution: first a hierarchical partition is obtained for the parameter space, by separate the space sequentially in subsequent directions and then define a distribution on this partition. The distribution is shown to be “point wise universal”, i.e. it achieved convergence in terms of entropy. The method is applied to several simulated and real datasets and compared to other methods through the log loss performance. Online computations are obtained by adapting the algorithm proposed for context tree switching. I have only minor comments. 1) The bibliography does not use the capital letters (example: AIC/BIC in reference 28) 2) Why is the truth stopping in Figure 2c? Given the different computational times (Figures are stopped at 30 minutes running time) they are of difficult interpretation. I think a better description in the text would be useful.

Reviewer 2



Originality: This seems like a new proposal for constructing a classifier based on a tree structure. Clarity: I found the paper very hard to read. A lot of the material is far from my usual field of research and I think that the authors are assuming a lot knowledge around ideas like Context Tree Switching. I found Figure 2 hard to read since the confidence intervals cross and some times cover other lines. Some lines are very short when plotted against n which makes them hard to interpret. The elapsed time goes to 1750 for a 30 minute run time. What are the units? To me it would clearer to plot NLL against elapsed time. You could then plot n against elapsed time to get an idea of the effects of computational speed. I found the description of the problem in 6.2 hard to understand and so it was hard to interpret the results. Significance: The paper seems like an interesting way to address problem but, since I'm familiar with a lot of methods in this area, it's difficult to give a clear opinion about significance.

Reviewer 3



This paper is unfortunately not in my field of expertise and while I have made significant efforts to understand it, I’m unsure if I should attribute the extreme difficulty I had in doing so to any shortcomings in the paper’s clarity or merely to my own lack of background. Several terms which I expected would be defined on first usage were not; this suggests either more effort is required to allow an educated but inexpert reader to understand the work, or that such readers are not the audience in the first place. I sincerely hope that the other reviewers are able to provide better informed assessments.

Reviewer 4



- The paper is sufficiently novel and presented clearly. A reader unfamiliar with the Context Tree Weighting technique might have a difficult first read, but as it is a well-known technique in information theory and its applications, I don't think this should be held against it. - The experimental comparison is somewhat light if the papers primary contribution was empirical, but I think it is sufficient in conjunction with the theoretical results, which are well done. Context Tree Weighting based variants have been used successfully in many different problems (data compression, bioinformatics), but typically deal with relatively low-dimensional binary side information, so this paper provides a method that fills this gap, and in my mind could be built on further by the machine learning community in the future. Some suggestions for future work: - There is a body of literature which improves on the KT estimator on sequences where the entire alphabet isn't observed, which is a common case when the data is recursively partitioned. I am quite certain the method advocated in this approach could benefit from applying the recently introduced SAD estimator (https://arxiv.org/abs/1305.3671) in place of the multi-KT, and the theoretical results extended to this case.

[Author Response · NeurIPS 2019]

**General.** We thank the reviewers for their detailed comments and suggestions. In order to address the remarks, Figure 2 will be replaced with the following one that uses a log scale on the x-axis for large $n$ cases to improve the readability. The zoom on GMM is no longer needed. We replace the plot of NLL vs elapsed-time for GMM by a comparison of $n$ vs elapsed time (in seconds) for GMM ($d = 2$) and GANMNIST ($d = 784$), as suggested by reviewer #2.

**Review #1** 1) We forgot to protect the capital letters in bibtex files. It will be corrected. 2) "Given the different computational times ... they are of difficult interpretation": we hope that the log scale on x-axis and the new figures relating $n$ and elapsed time will help the interpretation.

**Review #2** Model Switching is intuitively explained in the introduction (lines 39-42). We will add the term "switch": "*The resulting* **switch** *distribution still has a Bayesian form...*". Context Tree Switching builds on this idea and all the required definitions are included in this paper. As remarked by reviewer #4, Context Tree Weighting is a well-known technique in information theory and its applications and the necessary definitions are included in our paper. "I found the description of the problem in 6.2 hard to understand and so it was hard to interpret the results." We will add a reference to "Casella & Berger, Statistical Inference, Ch. 8".

**Review #3** See response to Review #2.

**Review #4** 1) "... more intuition regarding the use of the random data rotations.": random rotations are only used for the experiments on two sample testing to avoid unfair comparisons: without rotating the data, kd-trees easily find the direction featuring the differences, resulting in much higher performances – a fact illustrated by the following novel figure that will be included in the appendix.

(a) SG. $d = 50$.     (b) GMD. $d = 100$.     (c) GVD. $d = 50$.     (d) Blobs. $d = 2$.

2) Regarding the switching rate, the following remark will be added: *In [29], the authors observe better empirical performance with $\alpha_m^\gamma = n^{-1}$ for any cell $\gamma$, where $n$ is the number of samples observed at the root partition $\Omega$ when the $m$-th sample is observed in $\gamma$. With this switching rate they were able to provide a good redundancy bound for bounded depth trees. In our unbounded case, we observed a better empirical performance with $\alpha_m^\gamma = m^{-1}$.*

3) We will cite the suggested papers.

[Meta-Review · NeurIPS 2019]

This paper proposes a novel online algorithm for sequential probability prediction with side information. It combines nice theoretical results with sound experiments. While the paper is generally well written, I would encourage the authors to make the paper more accessible, by providing a bit more background on the different concepts introduced, either in the main paper or in the supplementary.